# Experimental Investigation of the Tribological Contact between Ti6Al4V-EBM Pin and UHMWPE Rotating Sheet for Prosthetic Applications

Annamaria Visco [1,2,*], Fabio Giudice [3], Eugenio Guglielmino [1], Cristina Scolaro [1] and Andrea Sili [1,*]

1   Department of Engineering, University of Messina, 98166 Messina, Italy
2   Institute for Polymers, Composites and Biomaterials, CNR-IPCB, 95126 Catania, Italy
3   Department of Civil Engineering and Architecture, University of Catania, 95123 Catania, Italy
*   Correspondence: avisco@unime.it (A.V.); asili@unime.it (A.S.)

**Abstract:** This work is aimed at studying the tribological contact between a titanium–aluminum–vanadium alloy pin (Ti6Al4V), produced by Electron Beam Melting (EBM), and a sheet of ultra high molecular weight polyethylene (UHMWPE), which are widely utilized materials for prosthetic applications. Using a "pin on disc" system, tribometric tests of different duration (up to 240 min) were carried out in order to trace the trend of the polymer mass loss as a function of test time. In this way it was possible to identify the stationary phase of adhesive friction, at which the specific wear rate, which characterizes the tribological system under different lubrication conditions, was obtained. As for the pin, no weight losses were measured, while the optical observations on the tip showed a compressive effect after the entire test campaign.

**Keywords:** Ti6Al4V; EBM; UHMWPE; tribometric test; wear

## 1. Introduction

The selection of materials capable of withstanding over time, and above all minimizing abrasions and stresses due to continuous tribological contacts, represents a challenge for research in the field of endoprostheses. With the purpose of ensuring mobility and load capacity, and also minimizing the problems of friction and wear, the use of biocompatible metal alloys, such as those based on titanium and the latest generation of polymers, has become widely established to produce prosthetic joints. Of course, they should be able to satisfy the demand for mechanical resistance and to guarantee, as much as possible, health safety needs [1]. Concerning titanium, and in particular its most widespread alloy Ti6Al4V, recent studies have highlighted the ergonomic efficiency of the components produced with the Electron Beam Melting (EBM) additive technique [2]. As known, one of the main applications of EBM is the production of biomedical implants customized for the individual patient, starting from computed tomography scans of the affected area [3].

Among the most suitable polymeric materials for prosthetic joints, ultra-high molecular weight polyethylene (UHMWPE) is considered, thanks to its excellent physical, chemical and mechanical characteristics, such as biocompatibility and good resistance to wear and abrasion, which can be increased by appropriate reinforcers as reported in the review by Macuvele et al. [4] and in more recent articles [5–7] concerning the use of carbon nanofiller and graphite. However, wearing contact with metal parts can lead to the release of debris, limiting the use of UHMWPE. Therefore, several experimental researches have been addressed to studying the improving effects on wear behavior due to lubrication [8] or to the development of suitable polymer composites [9]. In these works, the metal–polymer contact was reproduced by means of a "pin on disc" apparatus, in which a Ti6Al4V alloy pin, produced by EBM, acts through the pointed end on a polymer sheet, which is placed on a rotating disc. In these works, the wear tests were aimed at determining the volume

of material removed over a predetermined time interval. It allows a simple comparison between the different experimental conditions considered, such as the use of lubricants or the introduction of reinforcing fillers into the polymer. In particular, the best performances were obtained by carrying out the tests in simulated synovial fluid (SSF) and in bovine synovial fluid (BSF). However, this procedure does not allow documenting of the evolution of the wear process over time, nor the obtaining of the wear coefficient.

Concerning these latter aspects, it can be observed that the debris formation evolves from an initial transient phase with decreasing derivative to a stationary phase with a linear trend of the removed volume as a function of time. Thus, the wear coefficient, calculated at the stationary phase according to Archard's law, is the most suitable parameter to evaluate the behavior of a tribological couple [10]. In this regard, data relating only to some metal–metal systems [11] and ZrO2-polymer [12] are available.

Therefore, the main objective of the present work consists in studying how wear of the tribological pair Ti6Al4V/UHMWPE progresses over time, and characterizing the involved surface to achieve an accurate assessment of the phenomenon. The wear tests were carried out with a "pin on disc" tribometer with fixed metallic pin and movable polymer specimen as in a similar apparatus utilized by Tai et al., and Guezmil et al. [12,13]. The tribological behavior has been tested under different lubrication conditions, for increasing times up to 240 min, using a Ti6Al4V alloy pin, produced by EBM. The pointed end of the pin acts on UHMWPE specimens, produced by pressure-forming of powders at 200 °C, and placed on the rotating disk. The conditions of adhesive friction, which are generated on contact, led to the formation of grooves on the polymer surface and therefore to the production of debris. The identification of the stationary phase in the experimental curve of mass release as a function of test time allowed us to determine the specific wear rate. Consequently, the results obtained in the different lubrication conditions were compared.

Finally, the originality of this research work is given by the fact that in the literature there are no recent works on the tribology of the customizable Ti6Al4V alloy produced with EBM technology coupled with biomedical UHMWPE, to the best of our knowledge.

## 2. Materials and Methods

### 2.1. Production of the Ti6Al4V Alloy Pin

The pin used in the wear tests (length 15 mm and base side 10 mm) is provided with a rounded tip (therefore the metal contact surface can be considered spherical), and at the opposite end of a shank for its positioning (Figure 1).The pin was produced by Electron Beam Melting additive technology with the Arcam Q10 machine (Mölndal, Sweden), using Arcam AB powder (Mölndal, Sweden) (particle size between 45 and 100 µm) of the Ti6Al4V ELI-Grade 23 alloy with reduced content of O, N, C and Fe (Table 1). The low interstitial elements content allows for greater ductility and fracture toughness than Grade 5.

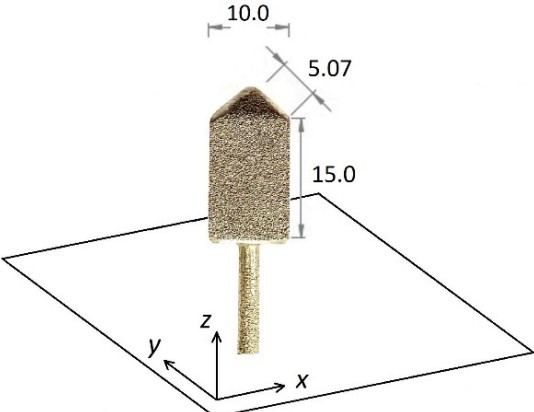

**Figure 1.** Set up of the pin with respect to the working plane x-y of the EBM process.

**Table 1.** Powder composition and limit of composition for Ti6Al4V ELI-Grade 23 alloy (weight %).

| Al | V | C | Fe | O | N | H | Ti |
|---|---|---|---|---|---|---|---|
| | | | Composition of the powder used | | | | |
| 6.0 | 4.0 | 0.03 | 0.1 | 0.1 | 0.001 | <0.003 | Bal. |
| | | Limits in composition range of the alloy Ti6Al4V ELI—Grade 23 (ASTM F136) | | | | | |
| 5.5–6.5 | 3.5–4.5 | <0.08 | <0.25 | <0.13 | <0.05 | <0.012 | Bal. |

Before melting, the powder was preheated at 650 °C to obtain a slight sintering, thus improving the electrical and thermal conductivity and the stability of the powder bed, also reducing temperature gradients and avoiding the occurrence of micro-fractures due to thermal variations [14]. The whole process was carried out in vacuum ($10^{-3}$ Pa), which is a particularly favorable condition for alloys of metals with a high affinity to oxygen such as titanium. During the selective melting phase, a small amount of inert helium gas at low pressure ($10^{-1}$ Pa) was added to avoid the accumulation of electrical charges in the powder and to ensure the process thermal stability, then the helium pressure was increased to facilitate cooling [15]. Figure 1 shows the positioning of the pin with respect to the process plane (x-y) and the direction of growth (z). Therefore, the pin cross-section is coplanar to the process plane and the longitudinal axis coincides with the layering direction.

The scanning method adopted for the beam is "snake x-y" type [16]: for each growth layer, the electron beam, suitably concentrated to obtain melting, impacts on the bed of pre-sintered powders, with a travel characterized by alternating parallel lines (serpentine); between one layer and the next, the direction of the alternating lines is rotated by 90°, so as to cross the previous layer. This type of scanning allows us to obtain components whose properties can be considered homogeneous with respect to the two orthogonal directions coinciding with the x-y axes of the process plane. The main process parameters values, shown in Table 2 [17], were selected according to the indications of the machine manufacturer [18], following the guidelines to reduce defects, such as surface roughness, presence of porosity, unfused regions, and delamination [19,20]. They correspond to a beam power of 900 W and a line energy of 200 J/m. In particular, the process parameters were set to ensure that the maximum depth of the molten pool slightly exceeds twice the thickness of the layer, and that the ratio between the thickness of the overlap between two adjacent layers and the thickness of the deposited layer remains in the optimal range 1.5–2.0 [21].

**Table 2.** EBM process parameters.

| Acceleration Voltage | 60 kV |
|---|---|
| Beam current intensity | 15 mA |
| Scanning speed | 4500 mm/s |
| Scan spacing | 200 μm |
| Layer thickness | 50 μm |

### 2.2. Production of UHMWPE Sheets

Samples of ultra-high molecular weight polyethylene (medical grade GUR1020-UHMWPE powder, average molecular weight of 2–4 × 106 g/mol, d = 0.93 g/cm$^3$ without calcium stearate, provided by Celanese Corporation, Irving, TX, USA) were obtained in the form of square plates (with a side of 20 mm and a thickness of 2 mm) by thermoforming the powder at 200 °C and increasing the pressure from 1 to 200 atm, with a holding time to maximum pressure equal to 16 min. A Teflon® sheet (0.1 mm thick) has been used for UHMWPE sheet releasing.

### 2.3. Tribometric Test

A prototype of the "pin on disc" device operating at room temperature was set up for the tribometric test (Figure 2a).

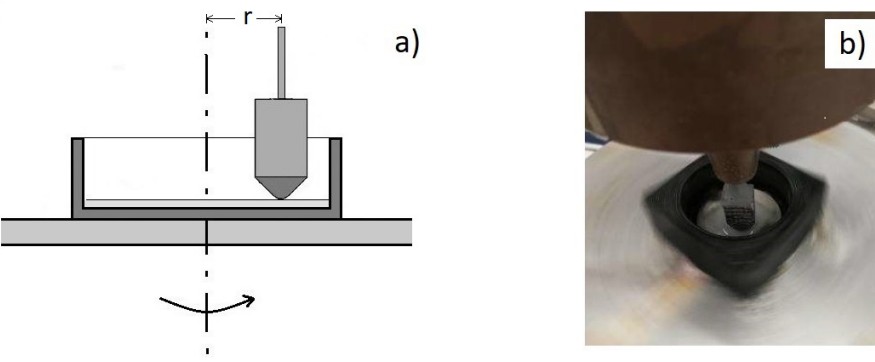

**Figure 2.** Sketch of the tribometer (**a**) and image of the pin in action on the polymer (**b**).

Wet conditions were obtained by depositing drops through a graduated pipette on the surface of the polymeric specimen. The latter is inserted in a containment tray screwed onto the rotating disc (Figure 2b). Distilled water and simulated synovial fluid (SSF) were used as lubricating fluids. The simulated synovial fluid (SSF) contained 0.3 wt % of Halyronic Acid (HA) in phosphate-buffered saline solution (pH 7.4). The inorganic electrolyte concentration in the SSF was: 153.1 mM of $Na^+$, 4.2 mM of $K^+$, 139.6 mM of $Cl^-$, and 9.6 mM of phosphate buffer.

Several works have considered polymeric pins acting on metal plates, but the opposite scheme is also present in literature: Tai et al. [12] and Guezmil et al. [13], refer to prosthetic joints with UHMWPE sheets placed on the movable disc. Moreover, the addition of fluids allows the contact to be kept submerged, thus maintaining the same lubricating condition during all the tests.

The disc rotates at frequency $\nu = 60$ round per minute (rpm) while the specimen is subjected to the action of the Ti6Al4V alloy pin with conical tip, whose tang is inserted in a rod that supports a weight of 30 N. The load of 30 N was constant in all the tribometric tests. The tribometric trials were carried out on various samples for different times up to 240 min. At each test the weight of the sample was measured to evaluate the mass loss of material and report the trend of this quantity as a function of time. For these measurements an electronic balance with sensitivity equal to $1 \times 10^{-4}$ g was used.

*2.4. Experimental Methods for Materials Characterization*

The composition of the pin was verified by a PAN analytical Epsilon 1 XRF spectrometer—(Malvern, UK). The pin microstructure was documented by digital optical microscopy observations (Hirox KH-8700 Digital Microscope—Tokyo, Japan) carried out on both the lateral surface and cross-section, as well as SEM observations and EDS measurements (JEOL JSM-7610F SEM-EDS apparatus—Tokyo, Japan). The observed surfaces were prepared with the usual metallographic methods and etched for 10 s using Kroll's reagent. Vickers microhardness tests were performed with a load of 500 g and a holding time of 15 s by the Future Tech. Corp. Vickers microhardness tester—Kanagawa, Japan.

On the surface of the polyethylene sheets, microhardness tests were carried out using a PCE-HT 210 Shore D durometer (PCE Deutschland GmbH, Meschede, Germany) and digital optical microscopy observations to document, at regular intervals, the wear tracks.

For the microhardness test, the average of ten measurements for each point of UHMWPE at different soaking times in SSF (0–15–60–240 min) was evaluated.

Through a three-dimensional analysis of the wear surfaces, detected by the Hirox KH-8700 Digital Microscope (Tokyo, Japan), it was possible to determine the surface roughness of the grooves, which was expressed by the average roughness ($R_a$) and the average roughness height ($R_z$) according to the JIS B0601 (2001) standard.

### 3. Results

#### 3.1. Characterization of the Ti6Al4V Alloy Pin

The Al content measured by XRF resulted in around 5%, less that of the powder. It is due to the Al volatility at the temperatures reached in the molten bath, as also found by other authors [22].

The morphology of the tip surface, shown by the rendering in Figure 3, appears grainy because of the EBM process modalities which involve powder particles with a size between 45 and 100 μm. The average roughness of the tip (calculated on 10 measurements) was: $R_a$ = 4.36 ± 0.51 μm and $R_z$ = 12.07 ± 1.05 μm.

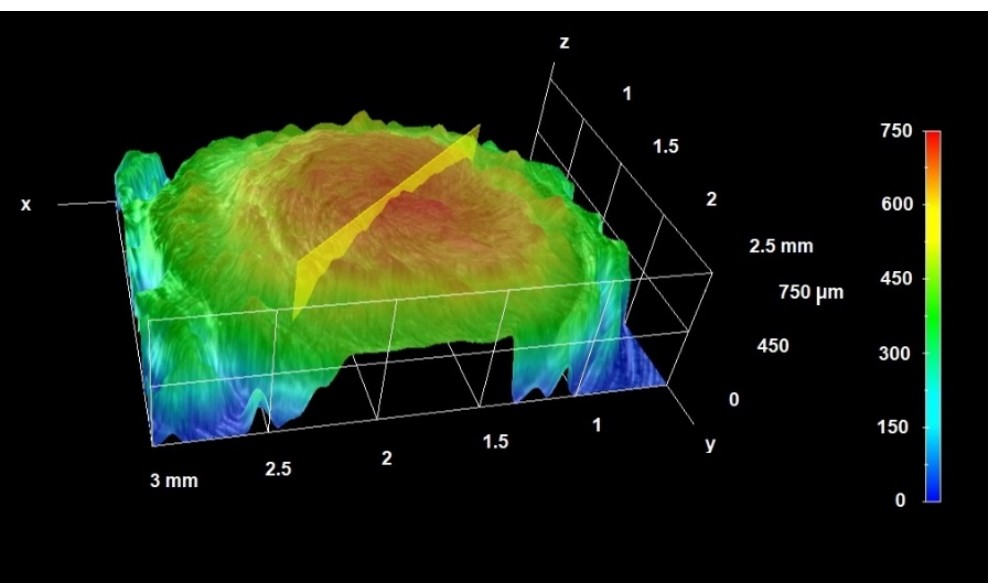

**Figure 3.** Tip rendering obtained by digital optical microscopy.

As known, the Ti6AL4V alloy solidifies in the β phase (centered cubic lattice), which remains stable above the β-transus temperature, in this case equal to about 980 °C, while the α phase (compact hexagonal lattice) is stable at lower temperatures. The allotropic transformation β → α is strongly conditioned by the presence of the two main alloying elements, aluminum (α-stabilizer) and vanadium (β-stabilizer) and by the cooling rate. Under the manufacturing process conditions, the microstructure at room temperature is lamellar biphasic (α-β), with the "basket weave" (Widmanstätten) morphology shown in Figure 4a,b referring respectively to the cross-section of the pin (plane x-y) and to a lateral face, therefore parallel to the direction of growth (z axis). Particularly, the latter case (Figure 4b) reveals the typical microstructure made up of columnar prior β grains delineated by the grain boundary α, and arranged along the direction of growth, due to the thermal gradient that exists along it [23,24]. Within prior β grains the α–β microstructure develops, with both lamellar colony and Widmanstätten morphologies.

The biphasic nature is confirmed by the EDS measurements reported alongside the SEM micrograph in Figure 5, which indicate a greater presence of vanadium in the inter-lamellar zones, therefore constituted by the β phase (spectrum 1), compared to the center of the lamellae in which the phase α is present (spectrum 2). The microhardness values give an average value on ten measurements equal to 319.8 ± 9.4 HV.

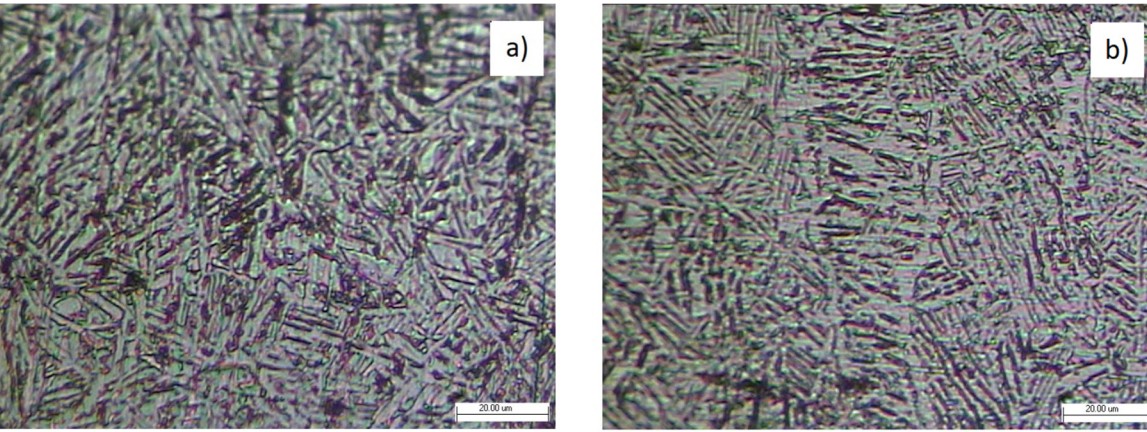

**Figure 4.** Optical micrographs of the pin: (**a**) cross-section; (**b**) lateral surface.

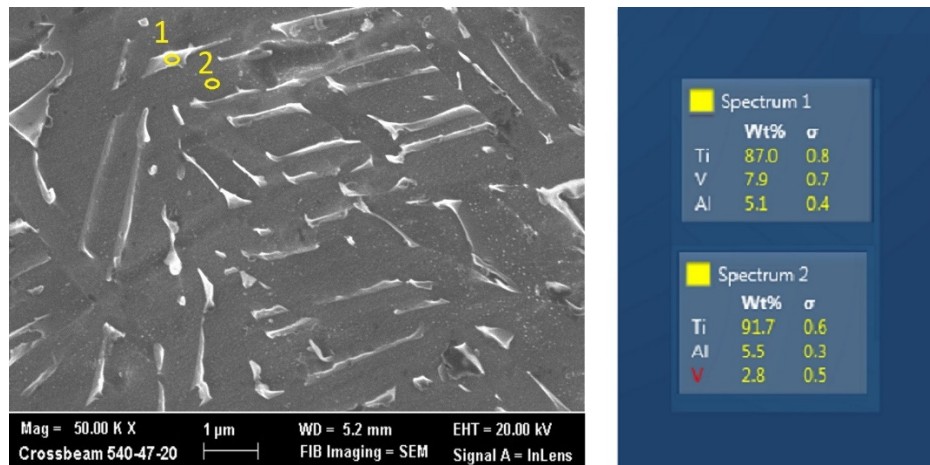

**Figure 5.** SEM micrograph and results of EDS measurements: interlamellar space (spectrum 1); inside a lamella (spectrum 2).

### 3.2. Tribometric Test

Figure 6 shows the results of the wear tests carried out with pure UHMWPE specimens for times (t) from 15 to 240 min, in the different lubrication conditions: dry, in water, and in simulated synovial fluid (SSF). In the three cases, due to the action of the metallic tip, polyethylene is subject to conditions of adhesive friction that lead to mass loss and to the formation of a circular groove on the sheet surface. On the other hand, as regards the Ti6Al4V alloy pin, no weight changes were detected.

It can be observed that the polymer mass loss ($\Delta m$) vs. time (t) curve presents an initial transient phase with a rapidly decreasing slope, followed by a linear stationary phase. This trend is confirmed by the interpolations of the experimental data through a markedly logarithmic curve in the first section (transient stage), which subsequently tend to be linear (stationary stage). These interpolating curves are represented for the three cases examined in Figure 6. The linearization of the stationary section was obtained by fitting the linear parameters on the experimental data, imposing the equality of the slopes at the intersection with the initial logarithmic section. The parameters of the interpolating curves, and the corresponding values of the determination coefficient $R^2$ are given in Table 3.

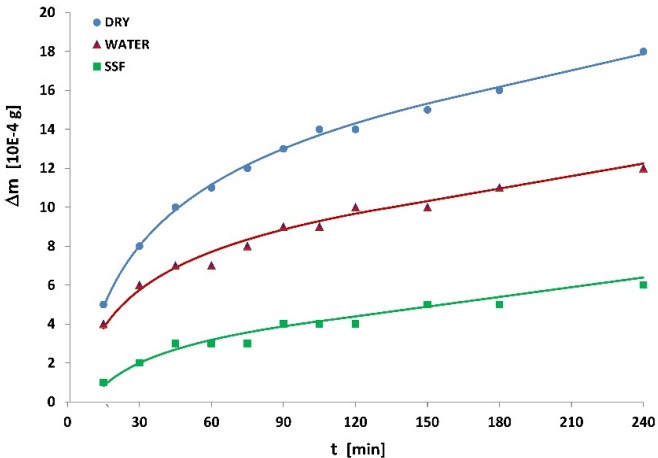

**Figure 6.** Results of the wear tests in the different lubrication conditions and interpolation of the experimental data: mass loss of the UHMWPE specimens (Δm) vs. test time (t).

**Table 3.** Interpolation of the experimental data: mathematical laws and parameters.

| Lubricating Condition | Transient Stage (Logarithmic Law) | | | Stationary Stage (Linear Law) | | |
|---|---|---|---|---|---|---|
| | $y = a_1 \cdot \ln(x) + b_1$ | | | $y = a_2 \cdot x + b_2$ | | |
| | $a_1$ | $b_1$ | $R^2$ | $a_2$ | $b_2$ | $R^2$ |
| Dry | 4.540 | 7.427 | 0.995 | 0.028 | 10.830 | 0.983 |
| Water | 2.829 | 3.876 | 0.978 | 0.021 | 6.928 | 0.974 |
| SSF | 1.713 | −3.831 | 0.965 | 0.017 | 1.950 | 0.992 |

*3.3. Investigations on the Contacting Surfaces*

The metal tip undergoes a morphological change during the wearing treatment, even in the case of lubrication with SSF (Figure 7). However, this variation is not such as to involve the removal of a significant amount of material. Rather, due to friction and localized overheating in the contact areas, as well as the weight of 30 N that the tip bears in the "pin on disc" system, there can be observed a redistribution of the metal mass that had been deposited during the EBM process. An estimate of the tip circumference, with respect to the positioning of reference particles, confirms an increase in diameter from approximately 3200 μm to 4300 μm. This demonstrates local compression and a widening effect on the tip end.

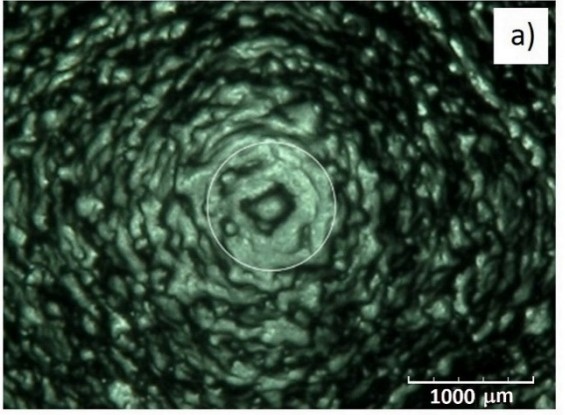
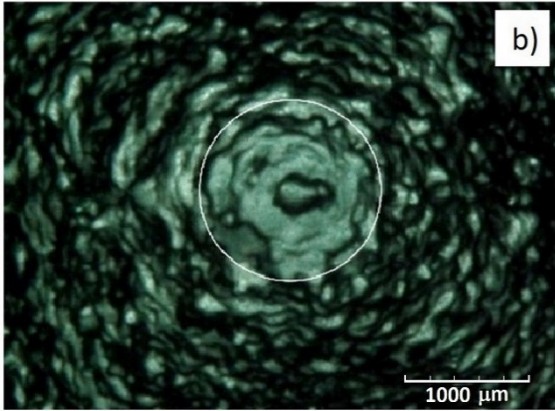

**Figure 7.** Ti6Al4V pin morphology before (**a**) and after (**b**) the wear test in SSF.

The UHWPE specimen's surface undergoes progressive damage during the wearing tests, through the formation of grooves or tracks of wear whose width initially increases very quickly, resulting in equal to 1244, 1332 and 1458 μm, respectively, after 15, 60 and

240 min (Figure 8a–c). The roughness (both $R_a$ and $R_z$) also increases with wear time: compared to the values measured after 15 min in SSF, after 240 min the $R_a$ value increases by 63.5% (from 2.44 μm to 3.99 μm), while $R_z$ increases by 74.2%, from 5.35 μm to 9.32 μm (Figure 8d–f). Furthermore, after 60 min of wear, the track loses its regular and continuous shape and damaged zones appear along the grooves.

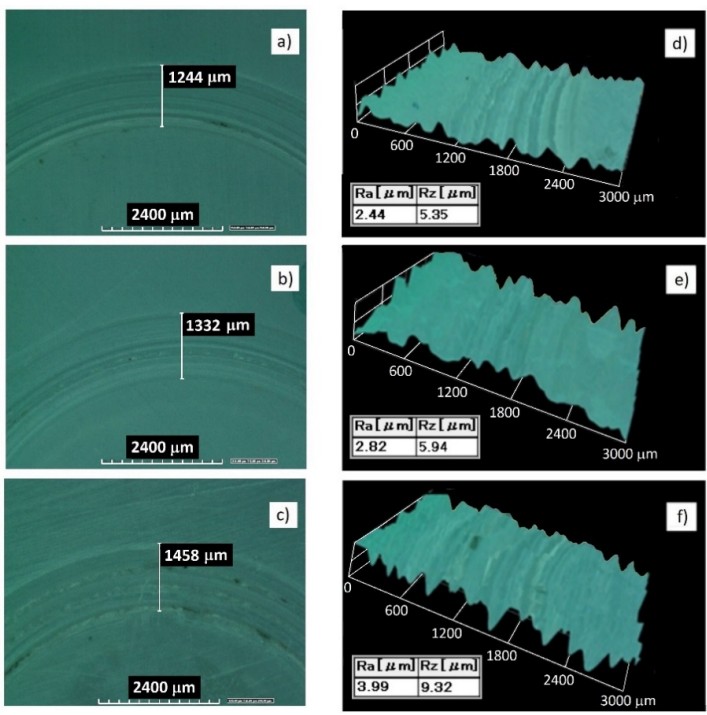

**Figure 8.** Optical observations of the polymer surface (**a**–**c**) and rendering of the wear tracks (**d**–**f**) at different test times: 15 min (**a**,**d**), 60 min (**b**,**e**) and 240 min (**c**,**f**).

The microhardness measurements, carried out in the wear tracks, show a softening of the polymer surface after wear under lubricating conditions with SSF, which is characterized by a tendency to gradually decrease as test time increases (Figure 9): in fact the initial value is reduced by about 2 HD in the first 15 min and by only 3 HD in the interval from 15 to 240 min. This suggests that the simulated synovial fluid, acting as a lubricant, is able to plasticize the surface of the polymer in contact with the tip, with a saturation effect as the contact with the tip persists.

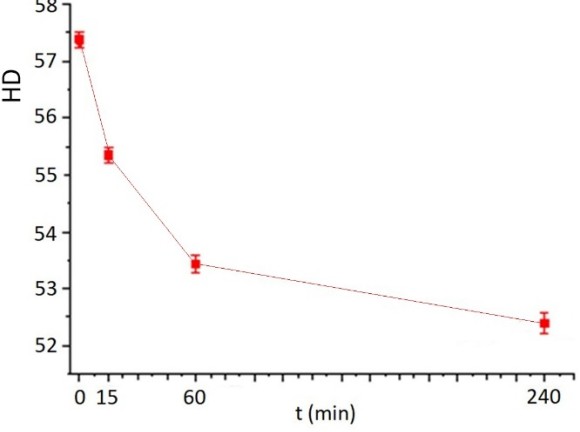

**Figure 9.** Shore D microhardness of UHMWPE at different soaking times in SSF.

## 4. Discussion

The material behavior under different contact conditions is often compared in terms of wear rate, which is defined as the ratio between the mass ($\Delta m$) or the volume (V) of the removed material and the exposure to wear, expressed by the time (t) or by the length of the sliding distance (L).

The experimental data of mass loss were therefore reworked to report the volume loss of material expressed by $V = \Delta m / \cdot (mm^3)$ (with $\rho = 9.355 \times 10^{-4}$ g/mm$^3$) as a function of the sliding length during the wear test $L = 2\pi r \cdot v \cdot t$ (m), where r (m) represents the radius of the circular track made by the tip on the specimen surface.

The curves of volume loss V vs. L are given in Figure 10: the transition from the initial to the stationary stage is marked with x. These diagrams show the possibility of committing a non-negligible mistake in the evaluation of the wear rate, if the contribution of the initial transient stage is not excluded. In fact the wear rate would be overestimated if not referred only to the stationary stage, being $V_{tot}/L_{tot} > V_s/L_s$.

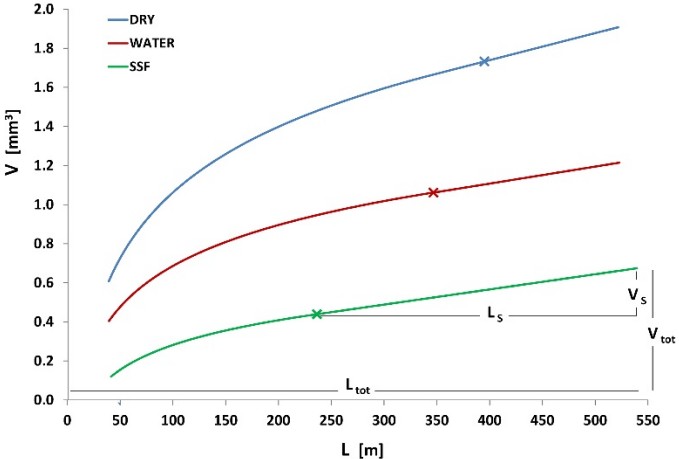

**Figure 10.** Volume loss (V) vs. the sliding distance (L) during the wear test under different lubricating conditions.

The trends of the respective derivatives (dV/dL) result in decreases as the sliding distance increases, until they assume a constant value (dV/dL)s when the stationary stage is reached (Figure 11).

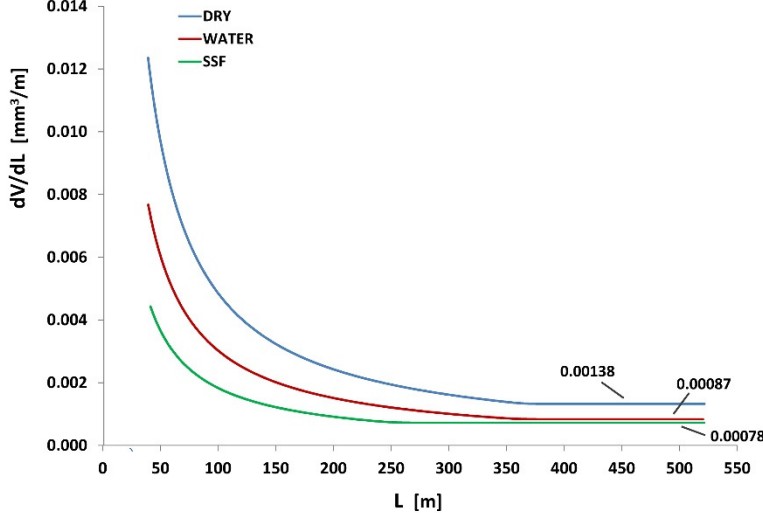

**Figure 11.** Derivative (dV/dL) vs. the sliding distance (L) during the wear test under different lubricating conditions.

For stationary conditions Archard's law can be applied [25]:

$$V = \frac{K \cdot P \cdot L}{H} \tag{1}$$

where V (mm$^3$) is the volume of the removed material, P (N) the applied load, L (m) is the sliding distance, and H is the hardness of the material that undergoes the wearing process.

The adimensional constant K = V·H/(P·L), known as the wearing coefficient, provides valuable indications for assessing the severity of wearing processes in different systems. In engineering applications, the quantity K/H = V/(P·L) (mm$^3$/(N·m)), defined as specific wear rate, is often more useful in cases where hardness H is difficult to evaluate [10]. To avoid errors in evaluation, both the wear coefficient and the specific wear rate must be referred to the stationary quantities and therefore the ratio V/L represents the slope of this stage, i.e., the derivative (dV/dL)s. The data in Table 4 show the specific wear rates for the contact Ti6Al4V alloy tip–UHMWPE sheet in the three different lubrication conditions, calculated as a ratio between the slope in the stationary stage (dV/dL)s and the applied load (P), differently from what was done in [8,9].

**Table 4.** Specific wear rates for the different contact conditions.

| Lubricating Condition | (dV/dL)s (mm$^3$/m) | P (N) | Specific Wear Rate (mm$^3$/N m) |
|:---:|:---:|:---:|:---:|
| Dry | 0.00138 | 30 | $4.6 \times 10^{-5}$ |
| Water | 0.00087 | 30 | $2.9 \times 10^{-5}$ |
| SSF | 0.00078 | 30 | $2.6 \times 10^{-5}$ |

As pointed out in [9], in dry conditions the wear track on the polymer sheet is deeper and particularly accentuated in the center. While in the case of lubrication with water, and even more evidently with the synovial fluid, the central wear groove becomes less and less deep. This leads us to think that, in the absence of lubricant, the mechanical action of the tip is mainly expressed through the formation of debris of polymeric material. This effect is demonstrated by the high wear in dry conditions and is particularly significant during the initial transient stage, especially when compared to the use of simulated synovial fluid as lubricant.

## 5. Conclusions

The effects of lubrication in the contact between the Ti6Al4V alloy tip and UHMWPE specimens were investigated by a "pin on disc" tribometer. By carrying out the wear tests at different times, an initial transient phase was observed in which the polymer mass loss as a function of time has a slope that rapidly decreases until a linear stationary behavior is reached. After the entire test campaign, the surface of the Ti6Al4V alloy tip shows a certain degree of deformation. However, no mass losses of the tip were measured. Nevertheless, more accurate future investigations will be aimed at evaluating the possible presence of metal traces, potentially dangerous for prosthetic joints, in the lubricating liquid. As for the polymer, the highest specific wear rate is observed in dry conditions, while this parameter is significantly reduced under lubricating conditions. The presence of the lubricant therefore reduces the action of the metal tip on the polymer sheet: hyaluronic acid, present in the simulated synovial fluid, confers greater lubricating power than distilled water; furthermore, this fluid produces a plasticizing effect on the surface of the polymer under the tip, as shown by the reduction of the hardness values as the test time progresses. Optical observations have shown how the degree of damaging on the polymer (track width, depth and roughness) increases with the test time, so much so that after 240 min severe zones of damaging are clearly evident.

In our work we used a spherical metal contact surface shaped with a smaller radius than those considered for prosthetic design as it facilitates the experimental tests. Therefore, the results presented here are the first step of a more in-depth future study on the

performance of Ti6Al4V alloy prostheses customized by the EBM technique, in which the tribological test will be performed with pins of more similar shapes to those present in the joints of prostheses.

**Author Contributions:** Conceptualization, F.G., A.V., E.G. and A.S.; methodology, F.G. and C.S.; validation, A.V. and A.S.; formal analysis, F.G. and C.S.; investigation, A.V., F.G., C.S. and A.S.; resources, E.G.; data curation, A.V., C.S. and A.S.; writing—original draft preparation, A.V. and A.S.; writing—review and editing, A.V., F.G., C.S. and A.S.; visualization, A.V., F.G., C.S., A.S. and E.G.; supervision A.V., F.G., E.G. and A.S. All authors have read and agreed to the published version of the manuscript.

**Funding:** This research received no external funding.

**Institutional Review Board Statement:** Not applicable.

**Informed Consent Statement:** Not applicable.

**Data Availability Statement:** The data presented in this study are available on request from the corresponding author.

**Acknowledgments:** Authors thanks Ing. Antonio Grasso, student of the Engineering Department of Messina University for the helpful discussion about the data analysis.

**Conflicts of Interest:** The authors declare no conflict of interest.

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
