# Peer review of "Experimental Investigation of the Tribological Contact between Ti6Al4V-EBM Pin and UHMWPE Rotating Sheet for Prosthetic Applications"

_metals, doi:10.3390/met12091526_

Round 1
Reviewer 1 Report
in the 1990’s many tests like these were completed by a broad range of companies / institutios and some fundamental observations were made. Firstly in the body the contact with the polymer is stationary, whilst the metallic surface articulates, thus the fresh smooth metal surface brings in / entrains fresh lubricant into the contact. Hence, the polymer is always the pin, and the metal is always the plate. Secondly, results dry or in water mean nothing as the test fluid must have biological proteins - usually this fluid is fetal calf serum, or actual synovial fluid. the constituents of the lubricant are crucial to prevent polymer transfer films and adverse thermal effects. The above 2 conditions dictate the fundamental wear processes within the contact.
the paper would be fine if you removed any links to joint replacements as the analysis is quite nice.
Author Response
Answer to the Reviewers by Visco et al.
ROUND #1
metals-1838659
Title
Experimental investigation on the tribological contact between Ti6Al4V-EBM pin and UHMWPE rotating sheet for prosthetic applications.
Reviewer 1
Comments and Suggestions for Authors
In the 1990’s many tests like these were completed by a broad range of companies / institutios and some fundamental observations were made.
About this consideration, we take the opportunity to clarify that the peculiarity of the present work consists in the study of wear occurring, under lubrication, between a Ti6Al4V conical pin with spheric contact surface manufactured by EBM and UHMWPE flat specimens. It is worth noting that the customizable artifacts produced by EBM have specific metallurgical and surface characteristics which distinguish them from cast or forged products.
Firstly in the body the contact with the polymer is stationary, whilst the metallic surface articulates, thus the fresh smooth metal surface brings in / entrains fresh lubricant into the contact. Hence, the polymer is always the pin, and the metal is always the plate.
Regarding the setup of the tribological test, several works considered polymeric pins acting on metal plates, but the opposite scheme is also present in literature (see the cited article of Tai et al. [ref.12] and the new added paper of Guezmil et al. [ref.13], which both refer to prosthetic joints and use a tribometer with fixed metallic pin and UHMWPE sheets placed on the movable disc). In any case, our method of adding the fluid allows the contact to be kept submerged, thus maintaining the same lubricating condition during all the tests.
Thus, we added these two sentences in the text:
- The wear tests were carried out with a “pin on disc” tribometer, with fixed metallic pin and movable polymer specimen as in a similar apparatus utilized by Tai et al., and Guezmil et al. [12,13]. (see section 1-Introduction, lines 63-64)
and
- “Several works considered polymeric pins acting on metal plates, but the opposite scheme is also present in literature: Tai et al. . [12] and Guezmil et al. [13], refer to prosthetic joints with UHMWPE sheets placed on the movable disc. Besides, the addition of fluids allows the contact to be kept submerged, thus maintaining the same lubricating condition during all the tests.” (see section 2.3, lines 134-138):
In our opinion, these results represent the first step of a more in-depth study on the performance of Ti6Al4V alloy for prostheses produced by EBM, in which the tribological test will be performed with pins of shapes more like those present in the articulations.
This was explained in the 5-conclusion section where we added this sentence:
“In our work we used a spherical metal contact surface shaped with a smaller radius than those considered for prosthetic design as it facilitates the experimental tests. Therefore, the results presented here are the first step of a more in-depth future study on the performance of Ti6Al4V alloy prostheses customized by the EBM technique, in which the tribological test will be performed with pins of more similar shapes to those present in the joints of prostheses. “(see lines: 320-325)
Secondly, results dry or in water mean nothing as the test fluid must have biological proteins - usually this fluid is fetal calf serum, or actual synovial fluid. the constituents of the lubricant are crucial to prevent polymer transfer films and adverse thermal effects. The above 2 conditions dictate the fundamental wear processes within the contact.
The tribological tests in dry and in water are carried out for comparison, being reference conditions commonly considered in literature.
We agree with the Reviewer about the crucial effects of the lubricant composition: in fact, as stated in the text, hyaluronic acid, present in the simulated synovial fluid, confers a greater lubricating power than distilled water; furthermore, this fluid produces a plasticizing effect on the surface of the polymer under the pin, as shown by the reduction of the hardness values ​​as the test time progresses. Tests with bovine fluid could be carried out in a future work to complete the present preliminary work.
the paper would be fine if you removed any links to joint replacements as the analysis is quite nice.
Authors thank the Reviewer for the very kind comment.
Reviewer 2 Report
Remarks
The submission metals-1838659 describes results of experimental investigation of the tribological contact between Ti6Al4V-EBM pin and UHMWPE rotating sheet for prosthetic applications. Such studies are interesting, however, the article needs some corrections:
1. The letter “n” is misunderstood in Title. Should it be “on” or “of”?
2. The Authors claim that the scheme “pin-on-disk” was used in the study, however, in reality it is a cone-on-disk scheme. Such shape of a contact area is not likely to be used frequently for prosthetic applications, and the Authors should justify in detail their decision,
3. The theoretical equations for calculating Ra and Rz are unnecessary in the article, since they are determined on different principles than integral equations; besides that, Rz should be calculated according to the ISO standard,
4. More detailed data regarding the SSF should be described,
5. Information about the load used (constant value) should be specified in the Methods section.
Author Response
Answer to the Reviewers by Visco et al.
ROUND #1
metals-1838659
Title
Experimental investigation on the tribological contact between Ti6Al4V-EBM pin and UHMWPE rotating sheet for prosthetic applications.
Reviewer 2
Comments and Suggestions for Authors
Remarks
The submission metals-1838659 describes results of experimental investigation of the tribological contact between Ti6Al4V-EBM pin and UHMWPE rotating sheet for prosthetic applications. Such studies are interesting, however, the article needs some corrections:
- The letter “n” is misunderstood in Title. Should it be “on” or “of”?
We agree with the reviewer observation and the title was corrected in “Experimental investigation of the tribological contact between Ti6Al4V-EBM pin and UHMWPE rotating sheet for prosthetic applications”.
- The Authors claim that the scheme “pin-on-disk” was used in the study, however, in reality it is a cone-on-disk scheme. Such shape of a contact area is not likely to be used frequently for prosthetic applications, and the Authors should justify in detail their decision
The pin consists of a conical body with a rounded tip; therefore, the contact surface is spheric, as specified in the revised text. Actually, the rounded tip was shaped with a radius smaller than those considered for prosthetic design since it facilitates the experimental tests. In any case the purpose of our work is to perform a basic investigation on the most common materials for biomedical applications verifying the effects of lubrication with SSF on wear behavior. Therefore, our results represent the first step of a more in-depth study on the performance of Ti6Al4V alloy prostheses customized by EBM, in which the tribological test will be performed with pins of shapes more similar to those present in the articulations.
Thus, we added the following sentences
- In section 2,1
…“therefore the metal contact surface can be considered spheric”.. (line 81)
- in section 5-Conclusions (pag.11/12, line 320-325):
“In our work we used a spherical metal contact surface shaped with a smaller radius than those considered for prosthetic design as it facilitates the experimental tests. Therefore, the results presented here are the first step of a more in-depth future study on the performance of Ti6Al4V alloy prostheses customized by the EBM technique, in which the tribological test will be performed with pins of more similar shapes to those present in the joints of prostheses. “
- The theoretical equations for calculating Ra and Rz are unnecessary in the article, since they are determined on different principles than integral equations; besides that, Rz should be calculated according to the ISO standard
Ra and Rz have been removed in the text (section 2.4) as suggested by the Referee, and we considered the JIS B0601 (2001) standard.
Thus, we rewrite the sentence at pag.4, line 163-166:“Through a three-dimensional analysis of the wear surfaces, detected by the Hirox KH-8700 Digital Microscope, it was possible to determine the surface roughness of the grooves, which was expressed by the average roughness (Ra) and the average rough-ness height (Rz) according to the JIS B0601 (2001) standard.”
- More detailed data regarding the SSF should be described.
We added this sentence in section 2.3, lines 131-133:
“Simulated Synovial Fluid (SSF) contains 0.3 wt % HA in phosphate buffered saline solution (pH 7.4). Inorganic electrolyte concentration in SSF was: 153.1 mM of Na+, 4.2 mM of K+, 139.6 mM of Cl-, an 9.6 mM of phosphate buffer.”
- Information about the load used (constant value) should be specified in the Methods section.4
As specified in paragraph 2.3, all the tests were carried out under a constant load equal to 30 N. To better specify this point, we added this sentence in the text (pag. 4 line 142): “The load of 30N was constant in all the tribometric tests.“
Authors are grateful to the Reviewers for the useful comments that improved the quality of their paper.

Reviewer 3 Report
The authors studied the wear of UHMWPE in different media by pin-on-disc test. I think the biggest shortcomings of this manuscript is the complete lack of novelty. Wear behaviour of UHMWPE has been extensively studied for decades. Even if the authors used a special home made Ti alloy pin, it does not change the well-known wear behaviour of PE. In addition, it has been also established that pure PE without any surface treatment shows the worst properties among PE based polymers. The authors reached the same conclusion in their former paper [ref 9]. It requires some explanation why the authors deemed worth studying more extensively a material having inferior behaviour. There is nothing any new among the conclusions reached comparing with that were reported in [9]. Moreover, credibility of the work is questioned considering that the authors reported a significantly different wear values for pure UHMWPE in their former paper [ref 9].